# New Species of *Mallocybe* and *Pseudosperma* from North China

**DOI:** 10.3390/jof8030256

**Published:** 2022-03-02

**Authors:** Ning Mao, Yu-Yan Xu, Tao-Yu Zhao, Jing-Chong Lv, Li Fan

**Affiliations:** College of Life Science, Capital Normal University, Xisanhuanbeilu 105, Haidian, Beijing 100048, China; 2210801013@cnu.edu.cn (N.M.); 2190801004@cnu.edu.cn (Y.-Y.X.); 2200802110@cnu.edu.cn (T.-Y.Z.); 2200802057@cnu.edu.cn (J.-C.L.)

**Keywords:** inocybaceae, multigene, phylogeny, taxonomy

## Abstract

Within the family Inocybaceae, many species of *Mallocybe* and *Pseudosperma* have been reported, but there are only a few reports on these two genera from north China. In this study, six collections of *Mallocybe* and 11 collections of *Pseudosperma* were studied by morphological and phylogenetic methods. Phylogenetic analyses based on sequence data from three or two different loci (ITS, LSU, and *rpb2* for *Mallocybe*; ITS and LSU for *Pseudosperma*) are performed to infer species relationships within genera *Mallocybe* and *Pseudosperma*, respectively. Results indicate that eight species of *Mallocybe* and *Pseudosperma* are found in Shanxi province, north China; two new species of *Mallocybe*, *M*. *depressa* and *M*. *picea*, are described. Overall, six species belong to *Pseudosperma*, of which three are new: *P*. *gilvum*, *P*. *laricis* and *P*. *pseudoniveivelatum*.

## 1. Introduction

Inocybaceae Jülich (Basidiomycota, Agaricales) is an ecologically important fungal family, and is estimated to contain 1050 species [1]. These ectomycorrhizal fungi of Inocybaceae form a mutually symbiotic association with as many as 23 families of vascular plants [2]. Inocybaceae was initially considered by many researchers to include only one or two genera [3,4,5,6,7,8,9]. Recently, Matheny et al. [2] revised Inocybaceae to include seven genera based on a six-locus phylogeny, namely *Auritella* Matheny & Bougher, *Inocybe* (Fr.) Fr., *Inosperma* (Kühner) Matheny & Esteve-Rav., *Mallocybe* (Kuyper) Matheny, Vizzini & Esteve-Rav., *Nothocybe* Matheny & K.P.D. Latha, *Pseudosperma* Matheny & Esteve-Rav., and *Tubariomyces* Esteve-Rav. & Matheny.

The genus *Mallocybe* was originally described as a subgenus of *Inocybe*. It is elevated to the genus level by Matheny et al. [2], with *Mallocybe terrigena* (Fr.) Matheny, Vizzini & Esteve-Rav. As the type species. The species of this genus are very widely distributed, reported in Africa, Asia, Australia, Europe, New Zealand, and North America [2]. Approximately 56 species are recorded in Index Fungorum [www.indexfungorum.org/Names/Names.asp (accessed on 5 February 2022)]. This genus is mainly characterized by a coarsely fibrillose or woolly-squamulose and often flattened pileus, which becomes noticeably dark upon the application of 5% potassium hydroxide; adnate lamellae; a short stipe; necropigmented basidia; short cheilocystidia (<50 μm long); and the absence pleurocystidia [2,7,10,11,12,13,14]. The genus *Pseudosperma* was originally included in *Inocybe* section *Rimosae* sensu stricto (= clade *Pseudosperma*) [15,16,17], and traditionally placed in the subgenus *Inosperma*. Now, it is one of the seven genera in Inocybaceae [2]. There are 93 records listed in Index Fungorum, and approximately 70 species are accepted according to Matheny et al. [2]. The species of this genus are characterized by fibrillose or rarely squamulose, often rimose pileus; furfuraceous to furfuraceous–fibrillose stipe, distinctly pruinose stipe apex; adnexed to sinuate lamellae; hyaline or not necropigmented basidia; cylindrical to clavate cheilocystidia; absent pleurocystidia; and spermatic odor [2,17,18,19].

During an investigation of Inocybaceae fungi in Shanxi (north China), some fruit-bodies of *Mallocybe* and *Pseudosperma* were collected. Subsequent morphological examination and molecular analyses showed they represented eight species, including five undescribed species. The aim of this study is to improve the knowledge of the genus *Pseudosperma* and *Mallocybe* by adding descriptions of five new species, and provide the DNA data of three previously described *Pseudosperma* species from China.

## 2. Materials and Methods

### 2.1. Morphological Studies

Collections were obtained and photographed in the field from Shanxi and Hebei province in China, dried in a fruit drier at 40–50 °C, and deposited in the herbarium of Capital Normal University, Beijing, China (BJTC) and the Herbarium Institute of Edible Fungi, Shanxi Academy of Agricultural Science, Taiyuan, China (HSA). Standardized color values were obtained from ColorHexa [http://www.colorhexa.com.asp (accessed on 5 February 2022)]. Microscopic characteristics were observed in sections obtained from dry specimens mounted in 3% KOH, Congo Red, or Melzer’s reagent [20]. The term ‘[n/m/p]’ means n. basidiospores from m. basidiomata of p collections. Dimensions of basidiospores are given using the following format ‘(a–)b–c(–d)’, where the range ‘b–c’ represents at least 90% of the measured values, and ‘a’ and ‘d’ are the most extreme values. L_m_ and W_m_ indicate the average basidiospore length and width (± standard deviation) for the measured basidiospore, respectively. ‘Q’ refers to the length/width ratio of basidiospores in side-view; ‘Q_av_’ refers to the average Q of all basidiospores ± standard deviation.

### 2.2. DNA Extraction, PCR Amplification, Sequencing

A small amount of basidiomata material (20−30 mg) was crushed by shaking for 45 s at 30 Hz 2−4 times (Mixer Mill MM301, Retsch, Haan, Germany) in a 1.5 mL tube, together with a 3 mm diam tungsten carbide ball. Total genomic DNA was extracted from the powdered basidiomata using NuClean Plant Genomic DNA Kit (CWBIO, Beijing, China), following the manufacturer’s instructions. Primers ITS1F and ITS4 were employed for the ITS [21,22], while LR0R and LR5 for LSU [23], and bRPB2-6F and bRPB2-7R2 for the *rpb2* were used [16,24]. Polymerase chain reactions (PCR) for the ITS region, LSU region, and *rpb2* gene were performed in 25 µL reaction containing 2 µL DNA template (concentration: 12−20 ng/µL), 1 µL primer (10 µM) each, 12.5 µL of 2 × Master Mix [Tiangen Biotech (Beijing) Co., Beijing, China], 8.5 µL ddH2O.

PCR reactions were implemented as follows: an initial denaturation at 94 °C for 5 min, then to 35 cycles of the following denaturation at 94 °C for 30 s, annealing at 52 °C for 45 s (ITS), 60 s (LSU and *rpb2*), 72 °C for 1 min; and a final extension at 72 °C for 10 min. The PCR products were sent to Beijing Zhongkexilin Biotechnology Co. Ltd. (Beijing, China) for purification and sequencing. The newly generated sequences were assembled and edited using SeqMan (DNA STAR package; DNAStar Inc., Madison, WI, USA) with generic-level identities for sequences confirmed via BLAST queries of GenBank. These sequences of *Mallocybe* and *Pseudosperma* were mainly selected from those used by previous studies [2,12,14,15,16,17,18,19,25,26,27,28,29,30,31,32]. The accession numbers of all sequences employed are provided in Appendix A.

### 2.3. Sequence Alignment and Phylogenetic Analyses

For this study, two datasets were assembled. Dataset I (ITS/LSU/*rpb2*) was used to investigate the phylogenetic placement of the *Mallocybe* species. *Pseudosperma triaciculare* Saba & Khalid and *P*. *breviterincarnatum* (D.E. Stuntz ex Kropp, Matheny & L.J. Hutchison) Matheny & Esteve-Rav. were selected as outgroup taxon. Dataset II (ITS/LSU) was used to investigate the phylogenetic placement of the *Pseudosperma* species. *Mallocybe velutina* Saba & Khalid and *M*. *africana* Aïgnon, Yorou & Ryberg were selected as outgroup taxon. The sequences of each marker were independently aligned in MAFFT v.7.110 [33] under default parameters, and edited by BioEdit 1.8.1. Maximum Likelihood (ML) and Bayesian Inference (BI) analyses were conducted on the resulting concatenated dataset.

Maximum Likelihood (ML) was performed using RAxML 8.0.14 [34] by running 1000 bootstrap replicates under the GTRGAMMAI model (for all partitions). Bayesian Inference (BI) analyses was performed with MrBayes v3.1.2 [35] based on the best substitution models (GTR + I + G for ITS and LSU; GTR + G for *rpb2*) determined by MrModeltest 2.3 [36]. A total of two independent runs with four Markov chains were conducted for 10 M generations under the default settings. Average standard deviations of split frequency (ASDSF) values were far lower than 0.01 at the end of the runs. Trees were sampled every 100 generations after burn-in (25% of trees were discarded as the burn-in phase of the analyses, set up well after convergence), and a 70% majority-rule consensus tree was constructed.

Trees were visualized with TreeView32 [37]. Bootstrap values (BS) ≥ 70% and Bayesian Posterior Probability values (BPP) ≥ 0.99 were considered significant [38,39].

## 3. Results

### 3.1. Phylogenetic Analyses

In this study, 36 sequences of ITS, LSU and *rpb2* were newly generated from our collections. Dataset I (ITS/LSU/*rpb2*) contained 122 sequences from 39 species, including 15 novel sequences of all three genes from our collections. *P*. *triaciculare* and *P*. *breviterincarnatum* were selected as the outgroup. The length of the aligned dataset was 2175 bp after exclusion of poorly aligned sites, with 620 bp for ITS, 884 bp for LSU, and 671 bp for *rpb2*. The topologies of ML and BI phylogenetic trees obtained in this dataset were practically the same, therefore only the tree inferred from the ML analyses is shown (Figure 1). The *Mallocybe* species formed a monophyletic lineage with strong support (MLB = 93%, BPP = 1.00). The sequences of our six collections formed two independent clades, which were respectively recognized and described as two new species: *Mallocybe depressa* and *Mallocybe picea*. *M*. *depressa* was sister to *M*. *velutina* Saba & Khalid with high supports, implying that they are closely related to each other. Another species *M*. *picea* was sister to *M*. *arthrocystis* (Kühner) Matheny & Esteve-Rav., with strong support (MLB = 96%, BPP = 1.00), and then grouped with *M*. *multispora* (Murrill) Matheny & Esteve-Rav. and *M*. *unicolor* (Peck) Matheny & Esteve-Rav. without supported data.

Dataset II (ITS/LSU) contained 1409 total characters (539 from ITS, 870 from LSU, gaps included) and included of 123 samples of 65 taxa. Since the topologies of ML phylogenetic trees is similar to that of the BI phylogenetic tree, only the tree inferred from the ML analyses is shown (Figure 2). A total of 21 sequences newly generated from our collections were resolved as six strong support clades, which indicated that they were six distinct species. Of them, the sequences of five collections clustered well with the authentic sequence of *Pseudosperma bulbosissimum* (Kühner) Matheny & Esteve-Rav., *P*. *rimosum* (Bull.) Matheny & Esteve-Rav., and *P*. *solare* Bandini, B. Oertel & U. Eberh., showing their identities with these three species, respectively. The remaining sequences of our collections formed three independent clades, which was recognized and described as three new species *Pseudosperma gilvum*, *Pseudosperma laricis,* and *Pseudosperma pseudoniveivelatum*. *P*. *gilvum* was sister to *P*. *citrinostipes* Y.G. Fan & W.J. Yu. *P*. *laricis* was closely grouped with *P*. *huginii* Bandini & U. Eberh and *P*. *arenicola* (R. Heim) Matheny & Esteve-Rav. with lower supports. *P*. *pseudoniveivelatum* was sister to *P*. *notodryinum* (Singer, I.J.A. Aguiar & Ivory) Matheny & Esteve-Rav.

### 3.2. Taxonomy

***Mallocybe depressa*** L. Fan, H. Zhou & N. Mao, sp. Nov. (Figure 3C and Figure 4)

MycoBank: MB843127

Diagnosis: *Mallocybe*
*depressa* is characterized by its golden yellow to yellowish-brown pileus, central depression of pileus when old, pileus margin splitting when mature, amygdaloid, and subamygdaloid to subcylindrical basidiospores, clavate to broadly clavate, and septate cheilocystidia, and usually grow in coniferous forest dominated by *Pinus* sp. It is most similar to *M*. *velutina,* but differs by its narrower basidiospores and broadly clavate cheilocystidia.

Etymology: *depressa*, refers to the depression in the center of the pileus with age.

Holotype: China. Shanxi Province, Taiyuan City, Xishan Forest Park, 37°82.39’ N, 112°46.99’ E, alt. 1100 m, 22 July 2021, on the ground in coniferous forest dominated by *Pinus* sp., J.Z. Cao CF1014 (BJTC FM1695).

Description—Pileus 10–35 mm wide, convex to plano-convex at young age, then applanates to uplifted, with a shallow depression at the center; margin initially decurved, becoming flattened and splitting with age; surface dry, strongly fibrous towards the margin, squamulose at the center, dark brown (#4e3000) around the disc, golden yellow (#ff9f00) to yellowish-brown (#cd7f00) elsewhere. Lamellae regular, adnate, brown (#915b25) to dark brown (#68421b) when mature, 1–2 tiers of lamellulae and concolorous with lamellae. Stipe 20–32 × 2.5–5 mm, central, equal with a slightly swollen apex and base, longitudinally fibrillose downwards the stipe, yellowish-brown (#cd7f00) to orange-brown (#9a4d00). Context pale yellow brown. Odor unrecorded.

Basidiospores [60/2/2] (7–)7.5–9(–11) × 4–5 μm, L_m_ × W_m_ = 8.33 (± 0.72) × 4.65 (± 0.34), Q = (1.4–)1.6–1.9(–2.2) (Q_av_ = 1.79 ± 0.16), smooth, amygdaloid, subamygdaloid to subcylindrical, sometimes ellipsoid, thick-walled, yellowish-brown. Basidia with yellowish necropigment, 20–30 × 6–8 μm, clavate, four-spored, occasionally two-spored. Cheilocystidia 15–36 × 9–12(–16) µm, often in clusters, septate, clavate to broadly clavate, occasionally balloon-shaped, apices rounded to obtuse, hyaline, thin-walled. Pleurocystidia absent. Caulocystidia only near the apex, 20–58 × 6–15 µm, clavate to elongate clavate, hyaline or pale yellow. Pileipellis a cutis, composed of parallel arranged of yellowish-brown to brown, cylindrical hyphae, often septate, 3–14 μm wide, thin-walled. Stipitipellis a cutis, composed of parallel, compactly arranged, hyaline, cylindrical hyphae, 4–12 μm wide, thin-walled. Clamp connections abundant in all tissues.

Habitat: In groups on the ground in coniferous forest dominated by *Pinus* sp., Hebei province and Shanxi province, China.

Additional specimens examined: China. Shanxi Province, Taiyuan City, Jinci Park, 38°57.18’ N, 113°30.52’ E, alt. 1370 m, 20 August 2020, on the ground in coniferous forest dominated by *Pinus* sp., H. Liu LH1266A (BJTC FM1300). Hebei Province, Chicheng county, Yanshan Mountains, 38°57.18’ N, 113°30.52’ E, alt. 947 m, 26 August 2020, on the ground in coniferous forest dominated by *Pinus* sp., H. Zhou 130732MFBPC643 (BJTC C643).

Notes: *Mallocybe velutina* is sister to *M*. *depressa* in our phylogenetic analyses (Figure 1). Morphologically, *M*. *velutina* differs from *M*. *depressa* by its pileus center fulvous, margin light yellow, larger, and broader basidiospores (9.0 × 5.4 µm on average) [13]. Molecular analyses reveal that *M*. *velutina* shares less than 96.04% similarity in ITS sequence with *M*. *depressa*, supporting their separation. *Inocybe caesariata* (Fr.) P. Karst. is recorded in Shanxi province, China [40]. It is similar to the new species by its pileus color and size. However, it can be differentiated from *M*. *depressa* by its pileus not splitting, white lamellae edge, and ellipsoid basidiospores. Another collected species *M*. *picea* is distinguished from *M*. *depressa* based on its larger basidiospores (10.44 × 5.69 µm on average) and broadly clavate to balloon-shaped cheilocystidia.

***Mallocybe picea*** L. Fan & N. Mao, sp. nov. (Figure 3A,B and Figure 5)

MycoBank: MB843129

Diagnosis: *Mallocybe picea* is characterized by its flattened and splitting pileus margin when mature, broadly clavate to balloon-shaped cheilocystidia, and usually grows in coniferous forest dominated by *Picea asperata*. It is most similar to *M*. *arthrocystis* but differs by its slightly broader basidiospores and often splitting pileus margin.

Etymology: *picea*, refers to the habitat of the species amongst forest of Picea.

Holotype: China. Shanxi Province, Wutai County, Wutai Mountain, 38°57.13′ N, 113°29.58′ E, alt. 2038 m, 25 July 2019, on the ground in coniferous forest dominated by *Picea asperata* Mast., L.J. Guo GLJM004 (BJTC FM555).

Description—Pileus 20–55 mm wide, hemispherical to broadly convex at young age, becoming plano-convex to applanate with age, with distinctly umbo; margin turned down when young, then flattened and splitting; surface dry, uniformly grayish velutinous when young, fibrillose to tomentose, earthy yellow (#e1a95f), yellowish-brown (#cd8526), sometimes dark brown (#764d16) towards the margin when mature. Lamellae regular, adnate, subdistant, yellowish-brown (#cc8526) to dark brown (#764d16), 2–3 tiers of lamellulae and concolorous with lamellae. Stipe 21–45 × 4–8 mm, hollow, central, equal, or sometimes slight widening at base, longitudinally fibrillose downwards the stipe, with white tomentose hyphae at the base, earthy yellow (#e1a95f) to yellowish-brown (#cd8526), paler white (#f5f5f5) at base. Context yellowish white. Odor unrecorded.

Basidiospores [100/2/3] (9–)9.5–11.5(–12) × (5–)5.2–6(–6.5) μm, L_m_ × W_m_ = 10.44 (±0.69) × 5.69 (±0.35), Q = 1.5–2.2 (Q_av_ = 1.84 ± 0.16), smooth, subamygdaloid to subcylindrical, cylindrical, thick-walled, yellowish-brown. Basidia with yellowish necropigment, (24–)28–39 × 8–10 μm, cylindrical to clavate, four-spored, rarely two-spored; sterigmata 2–5 μm long. Cheilocystidia 18–35 × 9–16 µm, often in clusters, septate, broadly clavate to balloon-shaped, apices rounded to subcapitate, hyaline, thin-walled. Pleurocystidia absent. Caulocystidia only near the apex, 22–50 × 7–10 µm, clavate to cylindric, hyaline or pale yellow. Pileipellis a cutis, composed of dense layers of repent hyphae; hyphae cylindrical, often septate, 6–15 μm wide and with yellowish-brown to brown intracellular or parietal pigment, thin-walled. Stipitipellis a cutis, made up of parallel, compactly arranged, thin-walled, cylindrical hyphae, 3.5–12 μm wide, hyaline or pale brown in KOH. Clamp connections abundant in all tissues.

Habitat: In groups on the ground in coniferous forest dominated by *Picea asperata*, Shanxi province, China.

Additional specimens examined: China. Shanxi Province, Wutai County, Wutai Mountain, 38°57.52′ N, 113°31.9’ E, alt. 1910 m, 25 July 2019, on the ground in coniferous forest dominated by *Picea asperata*, H. Liu LH636 (BJTC FM569). Ibid, 38°57.18’ N, 113°30.52′ E, alt. 2013 m, 27 August 2019, on the ground in coniferous forest dominated by *P*. *asperata*, Y. Shen SYM078 (BJTC FM896).

Notes: *Mallocybe picea* and *M*. *arthrocystis* are not only closely related phylogenetically, but also morphologically very similar. *Mallocybe arthrocystis* is originally reported from France and distinguished from *M*. *picea* by its pileus margin not splitting, slightly narrower basidiospores of 9.5–1.2 × 4.5–5.5 μm [12]. Molecular analyses also revealed that *M*. *arthrocystis* shares less than 90.23% similarity in ITS sequence with *M. picea*, supporting their separation. The species *Inocybe dulcamara* (Pers.) P. Kumm. is easily confused with *M*. *picea* in morphology, which is reported from China, but classified into *Mallocybe* by Fan and Tolgor [41]. *Inocybe dulcamara* differs from *M*. *picea* by its longer basidia 30–60 × 8–12 μm and white context in cap [12].

***Pseudosperma gilvum*** L. Fan & N. Mao, sp. nov. (Figure 3H,J and Figure 6)

MycoBank: MB843130

Diagnosis: *Pseudosperma gilvum* is characterized by convex to broadly convex pileus with subacute or obtuse umbo, mostly subphaseoliform, subcylindrical to cylindrical basidiospores, cylindrical, clavate to broadly clavate cheilocystidia. It is most similar to *P*. *triaciculare* but differs by its narrower basidiospores and paler pileus color.

Etymology: *gilvum*, Latin indicating light yellow, refers to the color of the pileus and stipe.

Holotype: China. Shanxi Province, Wenshui County, Lvliang Mountains, 37°28.28’ N, 111°34.22′ E, alt. 1750 m, 30 July 2021, on the ground in coniferous and broad-leaved mixed forest dominated by *Pinus* sp., L. FAN CF1115 (BJTC FM1941).

Description—Pileus 30–45 mm wide, convex to broadly convex with subacute or obtuse umbo; margin decurved or straight, not splitting; surface dry, fibrillose-rimulose, presence of a pale velipellis coating over the disc, light yellow (#ffffd4), becoming yellowish-brown (#ffc000) in some places with age, background pallid to cream white. Lamellae regular, adnate to sinuate, pale white (#f2f2f2) to grayish white (#e6e6e6) when young, becoming yellowish-brown (#ffb31a) with age, 1–2 tiers of lamellulae and concolorous with lamellae. Stipe 47–104 × 4–6 mm, solid, central, nearly terete, base slightly swollen, covered with whitish tomentum at young age, longitudinally fibrillose, pale yellow (#ffffd4) to yellowish brown (#ffab00), pale white at apex and base. Context white. Odor unrecorded.

Basidiospores [70/2/2] 10.5–12.5(–14) × (5.5–)6–7 μm, L_m_ × W_m_ = 11.40 (± 0.80) × 6.34 (±0.39), Q = 1.7–2.0 (Q_av_ = 1.80 ± 0.12), smooth, mostly subphaseoliform, subcylindrical to cylindrical, occasionally ellipsoid, slightly thick-walled, yellowish-brown to reddish-brown. Basidia 30–40 × 9–11 μm, clavate to broadly clavate, occasionally rounded-swollen at apex, primarily with four spored, rarely two spored, often with oily inclusions, hyaline in KOH. Cheilocystidia 25–75 × 9–14 µm, often in clusters, septate, cylindrical, clavate to broadly clavate, sometimes ovoid or subfusiform, often catenate with much shorter elements below the terminal element, hyaline to pale brown, thin-walled. Pleurocystidia absent. Caulocystidia only near the apex, 20–85 × 8–12 µm, clavate to cylindric, similar to cheilocystidia, hyaline or pale yellow. Pileipellis a cutis, composed of parallel, compactly arranged, thin-walled, hyaline or yellowish-brown, cylindrical hyphae, 4–12.5 μm wide. Stipitipellis a cutis, composed of compactly hyphae, 4–10 μm wide, hyaline or pale brown in KOH. Clamp connections abundant in all tissues.

Habitat: Scattered or in groups on the ground in mixed coniferous and broad-leaved forest dominated by *Pinus* sp., Shanxi Province, China.

Additional specimens examined: China. Shanxi Province, Pu County, Wulu Mountain, 36°33.34′ N, 113°30.52′ E, alt. 1910 m, 28 July 2021, on the ground in coniferous and broad-leaved mixed forest dominated by *Pinus* sp., N. Mao MNM275 (BJTC FM1875).

Notes: *Pseudosperma gilvum* is clustered with *P*. *citrinostipes* and *P*. *triaciculare* Saba & Khalid, and forms a distinct monophyletic group. This indicates that the three species are phylogenetically closely related to each other. However, there are clear differences in morphology among them. *Pseudosperma citrinostipes* has brownish yellow or straw yellow to golden yellow pileus, mostly ellipsoid spores, subfusiform, utriform to lageniform cheilocystidia and a different phylogenetic position (Figure 2) [19], that separates it well from our new species. *Pseudosperma triaciculare* can be distinguished by its darker pileus (brownish-orange to fulvous) with radially rimose margin, broader basidiospores (9.0 × 5.4 µm on average) and slightly smaller cheilocystidia (23–54 × 9–16 µm) [17]. Molecular analyses also reveal that *P. gilvum* shares less than 89.20% similarity in ITS sequence with *P*. *triaciculare*, supporting their separation. Both *P*. *brunneoumbonatum* Saba & Khalid and *P. gilvum* are presumed to be associated with *Pinus,* and have similar pileus shape. However, *P*. *brunneoumbonatum* differs from *P. gilvum* by its strongly rimose pileus margin and lager basidiospores (12.5 × 7.5 µm on average) [17]. We also collected some specimens of *P*. *bulbosissimum* (Kühner) Matheny & Esteve-Rav from Shanxi province, China. Its pileus is pale yellow, then ochraceous to reddish brown, and covered with fibrillose-rimose, similar to that of *P. gilvum*. The basidiospores of *P*. *bulbosissimum*, however, are remarkably larger (12–15 × 6–8 µm) [15].

***Pseudosperma laricis*** L. Fan & N. Mao, sp. nov. (Figure 3D and Figure 7)

MycoBank: MB843131

Diagnosis: *Pseudosperma laricis* is characterized by the pileus surface with fibrillose and strongly rimose, subcylindrical to cylindrical basidiospores and an ecological association with *Larix principis-rupprechtii*. It is most similar to *P*. *rimosum,* but differs by its narrower basidiospores and orange brown or brown pileus.

Etymology: *laricis*, refers to the habitat of the species amongst forest of *Larix*.

Holotype: China. Shanxi Province, Wutai County, Wutai Mountains, 38°57.7′ N, 113°30.16′ E, alt. 2075 m, 27 August 2019, on the ground in coniferous forest dominated by *Larix principis-rupprechtii* Mayr, Y. Shen SYM069 (BJTC FM887).

Description—Pileus 20–35 mm wide, convex, broadly convex or plane with subacute or obtuse umbo; margin at first incurved, then straight to somewhat wavy, not splitting; surface dry, smooth at the umbo, fibrillose and strongly rimose cracked towards center, yellowish-orange (#e6a800) to orange brown (#cc8400) or brown (#d18e4a), background pale white. Lamellae regular, adnate, grayish white (#e6e6e6) when young, becoming yellowish-brown (#ffb31a) to brown (#d18e4a) with age, 1–3 tiers of lamellulae and concolorous with lamellae. Stipe 51–65 × 4–6 mm, hollow, central, equal, longitudinally fibrillose, yellowish brown (#ffdf80) in different intensity, grayish white (#e6e6e6) at apex of the stipe. Context white. Odor unrecorded.

Basidiospores [85/2/2] (10–)11–14(–17) × (5–)5.5–7(–7.5) μm, L_m_ × W_m_ = 12.77 (± 1.38) × 6.16 (± 0.55), Q = 1.8–2.4 (Q_av_ = 2.07 ± 0.12), smooth, subcylindrical to cylindrical, slightly thick-walled, yellowish-brown to reddish-brown. Basidia 30–40 × 10–12 μm, clavate to broadly clavate, rounded-swollen at apex, generally with four spored, rarely two spored, often with oily inclusions, hyaline in KOH. Cheilocystidia 25–57 × 9–20 µm, often in clusters, mostly cylindrical, clavate to broadly clavate, rarely ovoid or fusiform, hyaline to pale brown, thin-walled. Pleurocystidia absent. Caulocystidia only near the apex, 17–47 × 8–11 µm, clavate or cylindrical, similar to cheilocystidia, often catenate with much shorter elements below the terminal element, hyaline or pale yellow. Pileipellis a cutis, composed of parallel, compactly arranged, thin-walled, yellowish-brown, cylindrical hyphae, with 4–13 μm wide. Stipitipellis a cutis, composed of parallel, compactly hyphae, 3–9 μm wide, hyaline or pale brown in KOH. Clamp connections abundant in all tissues.

Habitat: Scattered or in groups on the ground in coniferous forest dominated by *L*. *principis-rupprechtii*, Shanxi province, China.

Additional specimens examined: China. Shanxi Province, Wutai County, Wutai Mountains, 38°58.25’ N, 113°31.13’ E, alt. 1900 m, 27 August 2019, on the ground in coniferous forest dominated by *L*. *principis-rupprechtii*, H. Liu LH842 (BJTC FM924).

Notes: *Pseudosperma laricis* is clustered with *P*. *arenicola* (R. Heim) Matheny & Esteve-Rav. and *P*. *mimicum* (Massee) Matheny & Esteve-Rav. However, *P*. *arenicola* is distinguished by its stipe solid, often deeply buried in sand, and a broad host range, including species in *Salicaceae* and *Pinaceae*, and *P*. *mimicum* by its larger pileus (>65 mm) and ellipsoid basidiospores [15]. *Pseudosperma rimosum* (Bull.) Matheny & Esteve-Rav. is similar to *P*. *laricis,* and we also collected the fruit-bodies of *P. rimosum* from Shanxi Province, north China. It differs from the new species by its ellipsoid basidiospores (9.5–12.5 × 6–7 µm) [15]. Another species, *P*. *gilvum*, is easily distinguished from *P. laricis* by its paler color pileus and smaller basidiospores (11.4 × 6.34 µm on average).

***Pseudosperma pseudoniveivelatum*** L. Fan & N. Mao, sp. nov. (Figure 3E–G and Figure 8)

MycoBank: MB843132

Diagnosis: *Pseudosperma pseudoniveivelatum* is characterized by pileus surface with a distinct pale grayish to pale white velipellis, pileus margin splitting with age, ellipsoid basidiospores, broadly clavate to papillate or utriform cheilocystidia. It is most similar to *P*. *niveivelatum,* but differs by its smaller basidiospores and darker (yellowish-brown to brown) pileus.

Etymology: *pseudoniveivelatum*, refers to this species is similar to *P*. *niveivelatum*.

Holotype: China. Shanxi Province, Qinshui County, Lishan Mountains, 35°29.48′ N, 112°4.12′ E, alt. 1690 m, 7 July 2021, on the ground in coniferous and broad-leaved mixed forest dominated by *Quercus* sp., N. Mao MNM232 (BJTC FM1660).

Description—Pileus 20–45 mm wide, conical to conical-convex at first, then broadly convex to plane-convex with obtuse umbo; margin decurved or straight, becoming splitting with age; surface dry, with a distinct pale greyish to pale white velipellis, indistinctly fibrillose-rimulose, yellowish-brown (#cd9900) to brown (#8b4513), sometimes dark brown (#5e2f0d) at the center, background cream white. Lamellae regular, crowded, adnate, pale white (#ffffff) or yellowish white (#ffffe7) when young, later yellowish-brown (#ffbf00) to ochraceous (#a5682a), 1–2 tiers of lamellulae and concolorous with lamellae. Stipe 33–75 × 4–9 mm, solid, central, cylindrical, equal, or base slightly swollen, covered with whitish tomentum for a long time, later longitudinally fibrillose, pale orange (#ffae1a) to pale brownish (#997654). Context white. Odor unrecorded.

Basidiospores [70/2/2] (8.5–)9.5–11(–12) × (5–)5.5–7(–7.5) μm, L_m_ × W_m_ = 10.19 (± 0.77) × 6.22 (± 0.55), Q = 1.4–1.8 (Q_av_ = 1.63 ± 0.10), smooth, mostly ellipsoid, occasionally broadly ellipsoid or subcylindrical, slightly thick-walled, yellowish-brown to reddish-brown. Basidia 23–33 × 9–12.5 μm, clavate to broadly clavate, often rounded-swollen at apex, primarily four-spored, occasionally two-spored, usually with oily inclusions, hyaline in KOH. Cheilocystidia 30–65 × 7–16 µm, often in clusters, mostly clavate, broadly clavate to papillate or utriform, sometimes cylindrical, hyaline, thin-walled. Pleurocystidia absent. Caulocystidia only near the apex, 32–95 × 9–25 µm, in clusters, broadly clavate or utriform, at times with apices tapered or papillate, similar to cheilocystidia but larger, hyaline or pale yellow. Pileipellis a cutis, composed of parallel, compactly arranged, thin-walled, hyaline or yellowish-brown, cylindrical hyphae, 4–15 μm wide, with some encrustations, septate. Stipitipellis a cutis, composed of compactly hyphae, 5–14 μm wide, hyaline or pale brown in KOH. Clamp connections abundant in all tissues.

Habitat: Scattered or in groups on the ground in mixed coniferous and broad-leaved forests dominated by *Quercus* sp., north China, south China, and Europe.

Additional specimens examined: China. Shanxi Province, Qinshui County, Lishan Mountains, 35°29.14′ N, 112°1.20′ E, alt. 1660 m, 7 July 2021, on the ground in coniferous and broad-leaved mixed forest dominated by *Quercus* sp., N. Mao MNM224 (BJTC FM1656).

Notes: *Pseudosperma niveivelatum* is easily confused with *P*. *pseudoniveivelatum* in morphology, due to the presence of a white, abundant velipellis in *P*. *niveivelatum* that covers the pileus. However, this species has pale brown or yellow hues pileus, larger basidiospores (13.9 × 6.4 µm on average) and a different phylogenetic position (Figure 2) that separates it well from our new species [29]. *Pseudosperma notodryinum* is sister to *P*. *pseudoniveivelatum* in our phylogenetic analyses (Figure 2), implying that they have a close relationship. However, there are clear differences in the morphology. *P*. *notodryinum* can be distinguished by its darker pileus (yellow-ocher to rich yellowish-fuscous) and narrower basidiospores (9–12 × 5–6 µm) [42]. Molecular analyses revealed that *Pseudosperma notodryinum* shares less than 91.88% similarity in ITS sequence with *P. pseudoniveivelatum*, supporting their separation. A total of three species reported in China in previous studies, *P*. *obsoletum* (Quadr.) Valade, *P*. *perlatum* (Cooke) Matheny & Esteve-Rav. and *P*. *yunnanense* (T. Bau & Y.G. Fan) Matheny & Esteve-Rav., are all easily confused with *P. pseudoniveivelatum* in morphology [41,43]. However, *P*. *obsoletum* differs from *P. pseudoniveivelatum* by its gray brown to pinkish gray pileus, the absence of velipellis and narrower basidiospores (9–13 × 5–6 µm) [29]; it differs from *P*. *perlatum* by its larger basidiomata (pileus 35–100 mm, stipe 80–120 × 8–13 mm), and pileus color without yellow tinges [15]; and *P*. *yunnanens* by its fibrillose with densely squamules stipe and slightly smaller and narrower basidiospores (9–10.5 × 5–6 µm) [43]. Moreover, eleven ITS sequences respectively labelled ‘*P*. *obsoletum*‘, ‘*Inocybe obsoleta*‘ and ‘*P*. aff. *perlatum*‘ are conspecific to the new species *P. pseudoniveivelatum* since they clustered together with *P*. *pseudoniveivelatum* in ITS tree (not shown), and have more than 98.66% similarity in ITS region. Of them, two (MT072905, MG367271) are respectively from Inner Mongolia in northern China, Hainan Province in southern China, and nine from Europe (UDB035861, UDB015340, MW355002, MG367270, HG937630, JF908256, MZ410669, JX625280, JQ994477). These show that the new species *P. pseudoniveivelatum* is distributed in both northern and southern China and in Europe.

## 4. Discussion

Shanxi Province is located in north China, where the climate ranges from subtropical to cold temperate. Our analyses revealed two species of *Mallocybe* and six species of *Pseudosperma* in this region, i.e., *M*. *depressa*, *M*. *picea*, *P. bulbosissimum*, *P*. *gilvum*, *P*. *laricis*, *P*. *pseudoniveivelatum*, *P*. *rimosum* and *P. solare*. They all are associated with coniferous forests. *Pseudosperma bulbosissimum* is the most commonly encountered species, which is distributed in the central and northern regions in Shanxi Province. *Pseudosperma gilvum* is found in both central and southern regions. The remaining species probably have distribution limitations: *P*. *pseudoniveivelatum*, *P*. *rimosum,* and *P*. *solare* are distributed in the southern region, *M*. *depressa* are distributed in the central region, and *M*. *picea* and *P*. *laricis* are distributed in the northern region.

In China, the species diversity of the two genera of *Mallocybe* and *Pseudosperma* are scarce. A total of five species are reported in *Mallocybe* and 13 species in *Pseudosperma* [19,40,41,42,43,44,45,46]. With the exception of *P*. *citrinostipes*, *P*. *neoumbrinellum* (T. Bau & Y.G. Fan) Matheny & Esteve-Rav. and *P*. *yunnanense*, and the new species described in this study, the remaining seven species all need to be reexamined and verified with molecular data. They are *M*. *heimii* (Bon) Matheny & Esteve-Rav. [= *Inocybe heimii* Bon], *M*. *leucoloma* (Kühner) Matheny & Esteve-Rav. [= *Inocybe leucoloma* Kühner], *M*. *terrigena* (Fr.) Matheny, Vizzini & Esteve-Rav. [= *Inocybe terrigena* (Fr.) Kühner], *P*. *avellaneum* (Kobayasi) Matheny & Esteve-Rav. [= *I*. *avellanea* Kobayasi], *P*. *obsoletum* [= *I*. *obsoleta* Romagn.], *P*. *perlatum* [= *I*. *perlata* (Cooke) Sacc.], and *P*. *sororium* (Kauffman) Matheny & Esteve-Rav. [= *I*. *sororia* Kauffman] [19,40,41,42,43,44,45,46].

Key to the species of *Mallocybe* from China1. Annulus present*M*. *terrigena*1. Annulus absent22. Pileus applanate to uplifted, with a shallow depression at the right*M*. *depressa*2. Pileus plano-convex or applanate, with distinctly umbo or indistinctly umbo33. Habitat not associated with *Picea**M*. *heimii*3. Habitat associated with *Picea*44. Basidiospores subamygdaloid to subcylindrical, cylindrical, length > 9 μm*M*. *picea*4. Basidiospores ellipsoid to subphaseoliform, cylindrical, length < 9 μm*M*. *leucoloma*Key to the species of *Pseudosperma* from China1. Basidiomata uniformly brown (including pileus, lamellae, stipe)*P*. *neoumbrinellum*1. Basidiomata not uniformly brown22. Pileus color with pinkish tinges32. Pileus color without pinkish tinges43. Pileus yellowish buff, grayish brown to pale pinkish beige, basidiospores 10–15 × 5.5–7.5 μm*P*. *sororium*3. Pileus gray brown to pinkish gray, basidiospores 9–13 × 5–6 μm*P*. *obsoletum*4. Pileus surface with a distinct pale white or gray white velipellis54. Pileus surface without velipellis75. Stipe surface fibrillose with densely squamules*P*. *yunnanense*5. Stipe surface fibrillose without squamules66. Basidiospores mostly subphaseoliform, subcylindrical to cylindrical, L_m_ × W_m_ = 11.40 × 6.34 μm*P*. *gilvum*6. Basidiospores mostly ellipsoid, L_m_ × W_m_ = 10.19 × 6.22 μm*P*. *pseudoniveivelatum*7. Pileus color with yellow tinges87. Pileus color without yellow tinges98. Habitat associated with *Larix**P*. *laricis*8. Habitat not associated with *Larix*109. Pileus brownish to dark brown*P*. *perlatum*9. Pileus white or pallid ivory*P*. *bulbosissimum*10. Basidiospores small, length < 10 μm
*P. avellaneum*
10. Basidiospores large, length > 10 μm1111. Cheilocystidia missing (sub)capitate*P*. *rimosum*11. Cheilocystidia (sub)capitate1212. Stipe surface with lemon yellow fibrils*P*. *citrinostipes*12. Stipe surface with whitish to dingy whitish rough fibres or glabrous*P*. *solare*

## Figures and Tables

**Figure 1 jof-08-00256-f001:**
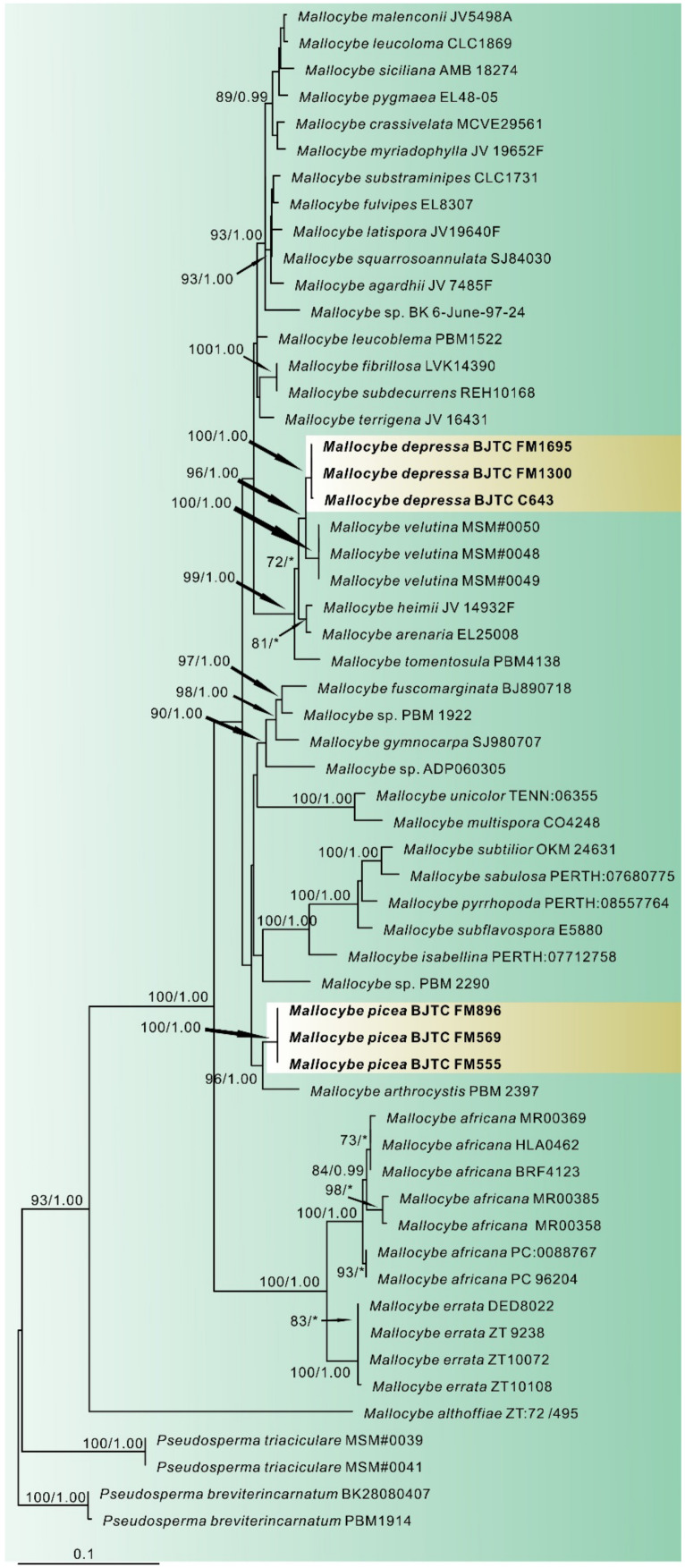
Phylogeny derived from Maximum Likelihood analysis of the combined (ITS/LSU/*rpb2*) dataset of *Mallocybe* and related genera in the family Inocybaceae. *Pseudosperma triaciculare* and *P*. *breviterincarnatum* were employed to root the tree as an outgroup. Numbers representing likelihood bootstrap support (BS ≥ 70%, left) and significant Bayesian posterior probability (BPP ≥ 0.99, right) are indicated above the nodes. New sequences are highlighted in black bold.

**Figure 2 jof-08-00256-f002:**
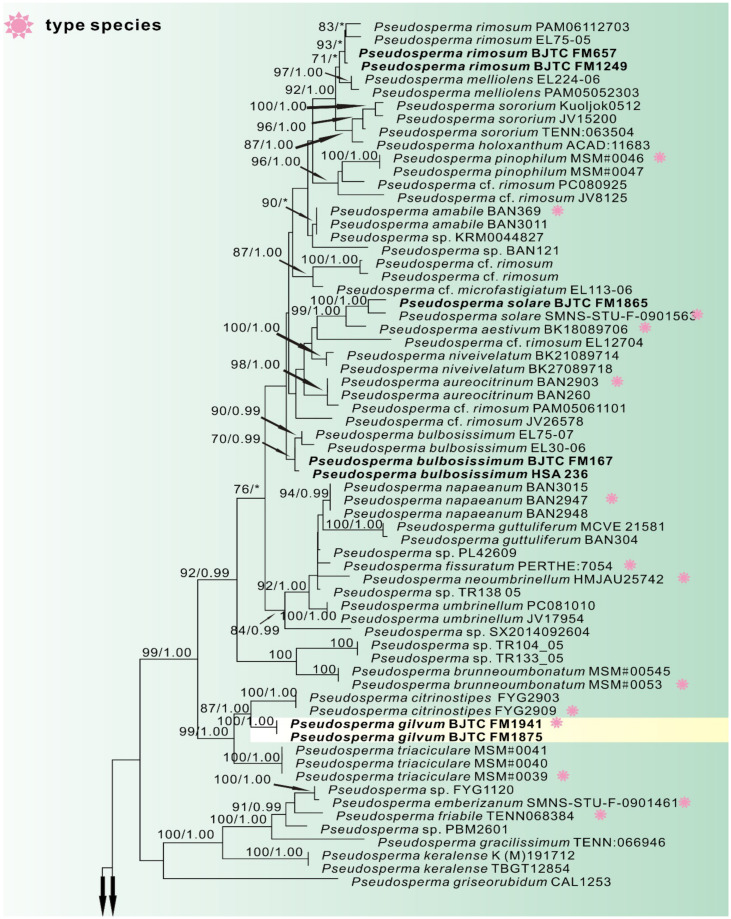
Phylogeny derived from Maximum Likelihood analysis of the ITS/LSU sequences from *Pseudosperma* and related genera in the family Inocybaceae. *Mallocybe velutina* and *M*. *africana* were employed to root the tree as an outgroup. Numbers representing likelihood bootstrap support (BS ≥ 70%, left) and significant Bayesian posterior probability (BPP ≥ 0.99, right) are indicated above the nodes. New sequences are highlighted in black bold. The red symbol represents the type species.

**Figure 3 jof-08-00256-f003:**
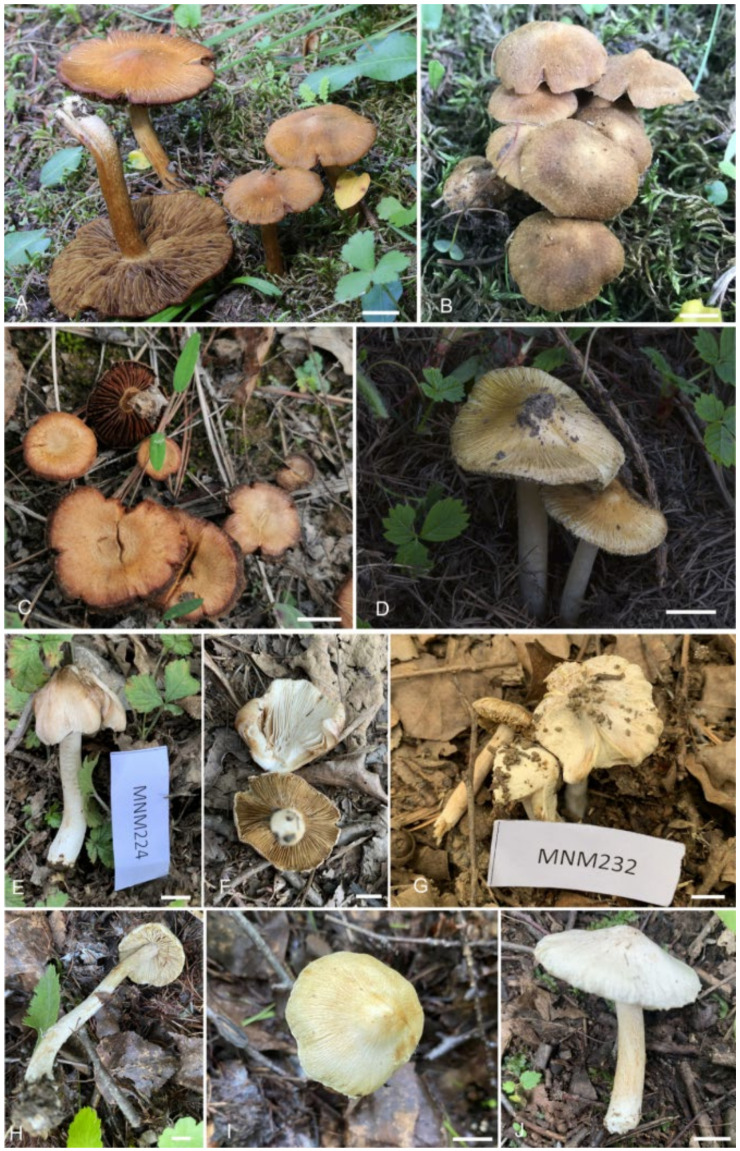
Basidiomata *of*
*Mallocybe* and *Pseudosperma*. (**A**,**B**). *M**allocybe picea* (BJTC FM555, holotype), (**C**) *Mallocybe depressa* (BJTC C643), (**D**) *Pseudosperma laricis* (BJTC FM887, holotype), (**E**–**G**) *Pseudosperma pseudoniveivelatum* (BJTC FM1660, holotype), (**H**–**J**). *Pseudosperma gilvum* (BJTC FM1941, holotype). Scale bars: (**A**–**J**) = 10 mm.

**Figure 4 jof-08-00256-f004:**
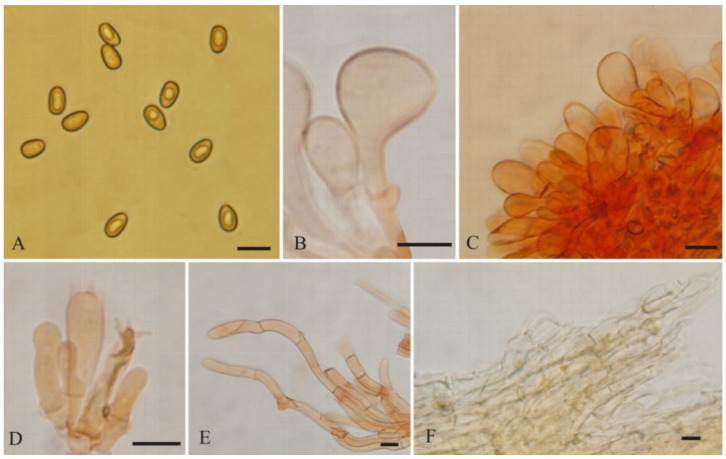
*Mallocybe depressa* (BJTC FM1695). (**A**) Basidiospores, (**B**,**C**) Cheilocystidia, (**D**) Basidia, (**E**) Caulocystidia, (**F**) Pileipellis. Scale bars: (**A**–**F**) = 10 μm.

**Figure 5 jof-08-00256-f005:**
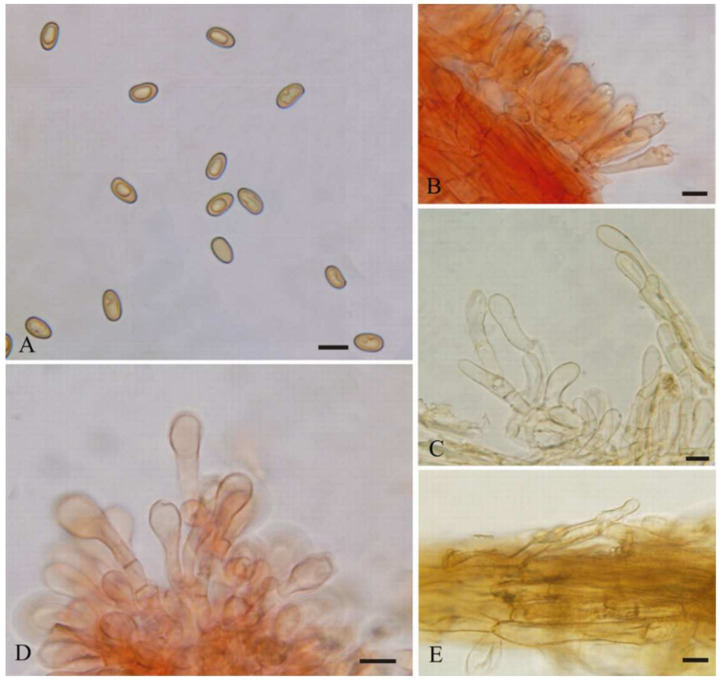
*Mallocybe picea* (BJTC FM555) (**A**) Basidiospores, (**B**) Basidia, (**C**) Caulocystidia, (**D**) Cheilocystidia, (**E**) Pileipellis. Scale bars: (**A**–**E**) = 10 μm.

**Figure 6 jof-08-00256-f006:**
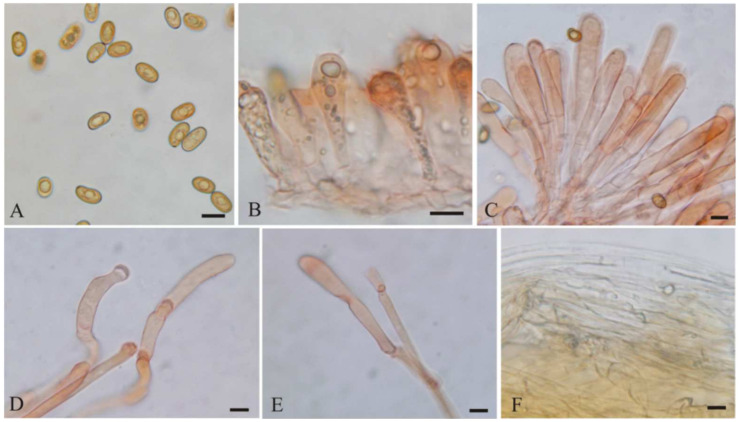
*Pseudosperma gilvum* (BJTC FM1941) (**A**) Basidiospores, (**B**) Basidia, (**C**) Cheilocystidia, (**D**,**E**). Caulocystidia, (**F**) Pileipellis. Scale bars: (**A**–**F**) = 10 μm.

**Figure 7 jof-08-00256-f007:**
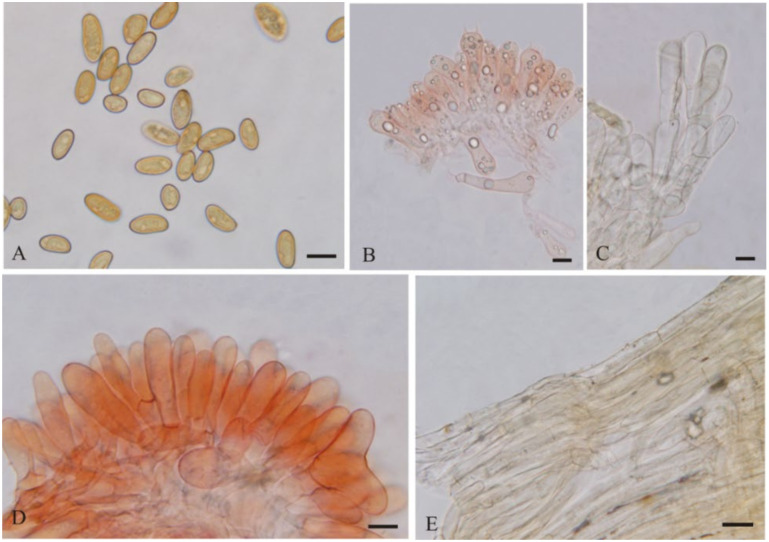
*Pseudosperma laricis* (BJTC FM887) (**A**) Basidiospores, (**B**) Basidia, (**C**) Caulocystidia, (**D**) Cheilocystidia, (**E**) Pileipellis. Scale bars: (**A**–**E**) = 10 μm.

**Figure 8 jof-08-00256-f008:**
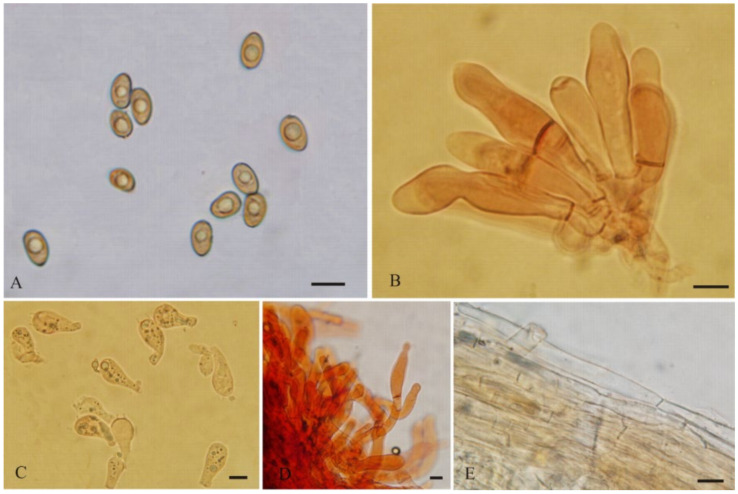
*Pseudosperma pseudoniveivelatum* (BJTC FM1660) (**A**) Basidiospores, (**B**) Cheilocystidia, (**C**) Basidia, (**D**) Caulocystidia, (**E**) Pileipellis. Scale bars: (**A**–**E**) = 10 μm.

## Data Availability

All sequence data are available in NCBI GenBank following the accession numbers in the manuscript.

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
