# Peer review of "New Species of Mallocybe and Pseudosperma from North China"

_jof, 2022, doi:10.3390/jof8030256_

Round 1

Reviewer 1 Report

This is a fine presentation of descriptions of several new taxa.

Most of my suggestions are editorial and can be found on the attached Word file.

Author Response

Thanks for your comments concerning our manuscript. Those comments are all valuable and very helpful for revising and improving our paper.

Point 1: Is this “ “ or ‘ ‘?

Response 1: Thank you very much. It is . I have made changes to this part of the entire manuscript. See Line 62 and 438.

Point 2: Surely these are not whole basidiomata that are crushed! How much?

Response 2: Thank you very much. It does have only a little bit of basidiomata material crushed. We describe this in more detail. See Line 70 [A small amount of basidiomata material (20−30 mg) was crushed].

Point 3: not sure what this means. That they have distinct ranges?

Response 3: Thanks for your question. Shanxi Province has a distinct climate difference between the south and the north. The southern part is warmer and the northern part is colder. At the same time, during the sample collection process in the past five years, these species can only be found in a specific range (other ranges are not found). Therefore, we think that these species may have limited distribution ranges.

Reviewer 2 Report

Dear Authours,

Please see my comments and suggestions in the attached file.

Please make sure all comments and suggestions are addressed before re-submit your manuscript. 

Well done!!!

Thanks

Author Response

Thanks for your comments concerning our manuscript. Those comments are all valuable and very helpful for revising and improving our paper. Your tips on drying temperature were very helpful for
my follow-up research.

Point 1: diagnosis should include all the specific characteristics of this species that can be distinguish from other species. The depression of the pileus when it is old is a unique characteristic of this species but it is not included in the diagnosis.

Response 1: Thank you very much for your suggestion. Your reminder is very helpful to me. We have added this characteristic to the diagnosis section.See Line 169 (central depression of pileus when
old.)

Point 2: please add a table to compare your species with these species.

Response 2: Thanks for your suggestion. We have added the species key to the manuscript. it's below the discussion.

Reviewer 3 Report

Dear Authors

Description of five new species of Mallocybe and Pseudosperma in North China, presented by the authors seems interesting to me. The manuscript overall is good. The content of the paper could be published in JoF. In the manuscript, I pointed out a few minor corrections.

Best regards

Author Response

Thanks for your comments concerning our manuscript. Those comments are all valuable and very helpful for revising and improving our paper.

Point 1: in what concentration?

Response 1: Thank you very much. We describe this in more detail. See Line 77 [LSU region and rpb2
gene were performed in 25 μ L reaction containing 2 μ L DNA tem-plate (concentration: 12−20 ng/μL)].

Point 2: Why is the RPB2 gene not used?

Response 2: Thanks for your question. When we go to study the phylogenetic relationship of the genus Pseudosperma, we hope to accommodate the molecular data of more species. However, when
we sorted out these molecular data, we found that the genes of rpb2 were very few compared to ITS and LSU, and the coverage was very limited. At the same time, based on the research of Bandinis et al. and Saba et al., ITS and LSU genes can well explain the phylogenetic relationship.

All best
